# Combination Treatment for Inhibition of the Growth of *Staphylococcus aureus* with Recombinant SAP8 Endolysin and Nisin

**DOI:** 10.3390/antibiotics11091185

**Published:** 2022-09-02

**Authors:** Seon-Gyu Kim, Shehzad Abid Khan, Young-Duck Lee, Jong-Hyun Park, Gi-Seong Moon

**Affiliations:** 1Major of Biotechnology, Korea National University of Transportation, Jeungpyeong 27909, Korea; 24D Convergence Technology Institute, Korea National University of Transportation, Jeungpyeong 27909, Korea; 3Department of Food Science and Engineering, Seowon University, Cheongju 28674, Korea; 4Department of Food Science and Biotechnology, Gachon University, Seongnam 13120, Korea

**Keywords:** *Staphylococcus aureus*, recombinant SAP8 endolysin, nisin, combination effect

## Abstract

*Staphylococcus aureus*, a pathogenic species of genus *Staphylococcus* involved in foodborne illness always remain among the top priorities of the world major concerns. In the present study, we have used recombinant SAP8 endolysin from the bacteriophage SAP8 and commercial nisin to inhibit the viability of pathogenic *S. aureus* KCTC 3881 cells; however, the approach was not identified as cost-effective. A gradual decrease in the viable *S. aureus* KCTC 3881 cell counts was observed with an increase in the concentrations of recombinant SAP8 endolysin and nisin. However, combined treatment with recombinant SAP8 endolysin and nisin decreased the viable *S. aureus* KCTC 3881 cell counts in a significant manner. The combination of 0.01 µM of recombinant SAP8 endolysin with 9 IU/mL and 18 IU/mL of nisin demonstrated a promising decrease in the viable cell counts of the strain. Under the scanning electron microscope, the combination treatment with 0.01 µM of recombinant SAP8 endolysin and 18 IU/mL of nisin showed complete cellular destruction of *S. aureus* KCTC 3881. We propose that a combination of recombinant SAP8 endolysin and nisin could be a strong alternative to antibiotics to control the growth of *S. aureus* including MRSA.

## 1. Introduction

Food safety always remains among the top priorities of the world major concerns. The global burden due to foodborne illnesses and their economic and social impacts remains excessively high [1]. The foodborne diseases caused by pathogenic gram-positive and gram-negative bacteria always remains a serious concern globally. *Staphylococcus* (*S.*) *aureus* is a Gram-positive, round-shaped, facultatively anaerobic, ubiquitous microorganism, and considered as an important foodborne pathogenic species of genus *Staphylococcus* [2,3]. *S. aureus* persist in food setting, form biofilm, release enterotoxins in food, show resistance to typical antimicrobials, and is the third major cause of foodborne illness in humans [4,5]. The antibiotic therapies against *S. aureus* are not showing satisfactory results, therefore development of new anti-bacterial strategies against these highly antibiotic resistance bacteria is strongly needed. Resistance of *S. aureus* to multiple antibiotics has urged the employment of bacteriophages (bacterial viruses) as an alternative option of therapies for the treatment of biofilm formation and infections [6]. Food and Drug Administration (FDA) have approved bacteriophages and nisin as food additives to control pathogenic bacteria and their toxins [7,8,9]. Duc et al. have used phage SA46-CTH2 and nisin together to combat *S. aureus* [10]. Although phage therapy has been successfully employed against pathogenic bacteria, concern exists about the safety and development of bacterial resistance against phage [11]. Hussain et al. have studied about the marine *Vibrio* resistance mechanism against phages [12,13]. However, in the recent past, bacteriophage endolysins have been successfully employed against several pathogenic bacteria [14,15,16]. Yu et al. examined endolysin SAP8 activity against MRSA and found it as an alternative agent to control *S. aureus* strains [16]. In comparison with bacteriophage endolysins, bacteriocins from lactic acid bacteria have been successfully extracted and employed to control the growth of food-borne pathogenic bacteria. Nisin, a commercialized product, extracted and purified from *Lactococcus lactis*, has been used as a bio-preservative in the food industry [17]. In previous studies, nisin has successfully been employed against *S. aureus*, and significant reduction in growth of *S. aureus* was observed [18]. While the antibacterial effects of bacteriophage endolysin SAP8 and nisin have been studied previously, the combination or synergistic effects are still unexplored.

In this study, we have investigated the effects of recombinant SAP8 endolysin from the bacteriophage SAP8 and commercial nisin in controlling the growth of *S. aureus* KCTC 3881. Furthermore, the combination effect of recombinant SAP8 endolysin and nisin against the strain has been studied.

## 2. Materials and Methods

### 2.1. Antimicrobial Activity of the Recombinant SAP8 Endolysin against Staphylococcus aureus

The recombinant DNA for SAP8 endolysin was previously constructed and expressed in *Escherichia coli* strain and the recombinant protein named LysSAP8 was named recombinant SAP8 endolysin in this study [16]. The antimicrobial activity of the recombinant SAP8 endolysin mentioned above was examined against methicillin resistant *S. aureus* (MRSA) KCTC 3881 isolated from human lesion. Previously, 73 strains of *S. aureus*, including MRSA strains, were found to be susceptible against recombinant SAP8 endolysin, including *S. aureus* KCTC 3881, which demonstrated that recombinant SAP8 is not strain-specific [16]. In our previous study, we tested crude nisin from *L. lactis* strain against several *S. aureus* strains and found strongest activity of nisin against *S. aureus* KCTC 3881, chosen as a model strain in this study [19]. To test activities of recombinant SAP8 endolysin and nisin, *S. aureus* KCTC 3881 strain was cultured in MRS broth (BD, Sparks, MD, USA) at 37 ℃ for 12 h. To measure CFU/mL, 10 times serial dilutions of culture was made, then we poured 10 µL of each dilution on MRS agar plate in triplicate form. After incubation at 37 ℃ for 24 h, colonies were counted and determined CFU/mL using formula (total colonies × dilution factor × 100). The CFU/mL of broth culture medium appeared 10^9^ CFU/mL, 1 mL of culture broth was taken, and centrifuged at 2451× *g* for 10 min at 4 ℃. The cell pellet was washed and resuspended with distilled water and diluted to attain approximately 10^8^ CFU/mL of cells. Ten-fold serial concentrations of the recombinant SAP8 endolysin were added to distilled water inoculated with 10^6^ CFU/mL of *S. aureus* KCTC 3881 cells. The mixture was incubated at 30 ℃ for 3 h and viable cell counts were measured.

### 2.2. Co-Treatment with the Recombinant SAP8 Endolysin and Nisin against Staphylococcus aureus

*S. aureus* KCTC 3881 cells were prepared following the above-mentioned method. The cell pellet was washed and resuspended with distilled water and diluted to attain approximately 10^6^ CFU/mL of cells. The recombinant SAP8 endolysin (0.01 µM) and nisin (9 IU/mL and 18 IU/mL; Galacin Nisin 101, Galactic, Belgium) were added simultaneously to distilled water inoculated with 10^6^ CFU/mL of S*. aureus* KCTC 3881 cells. The mixture was incubated at 30 ℃ for 3 h and viable cell counts were measured at times. To calculate minimum inhibitory concentration (MIC) of SAP8 endolysin, nisin, and a combination of SAP8 endolysin and nisin, resazurin assay was performed as described by the manufacturer, with slight modifications [20]. Briefly, in 96 well microplates, 20 µL of 10^6^ CFU/mL of *S. aureus* was inoculated in 176 µL of MRS broth medium and 4 µL of different concentrations of the recombinant SAP8 endolysin (0.1 µM, 1 µM, 5 µM, 10 µM), nisin (9 IU/mL, 18 IU/mL, 36 IU/mL, 72 IU/mL), and in combination with 0.01 µM of SAP8 endolysin were added in triplicates. The microplate was incubated at 37 ℃ for 12 h. After incubation, 20 µL of resazurin (1000 ppm) were added to each well of microplate and incubated in an orbital shaker at 37 ℃ for 4 h at 100 rpm for resazurin metabolization.

### 2.3. Scanning Electron Microscopy of S. aureus Co-Treated with Endolysin and Nisin

As mentioned above, 20 µL of an aliquot of *S. aureus* KCTC 3881 cells after 3 h of co-treatment with 0.01 µM of the recombinant SAP8 endolysin and 18 IU/mL of nisin were taken and centrifuged to remove the culture supernatant. After centrifugation, 20 µL of 50% of EtOH was added to the cell pellet, the cells were resuspended and incubated for 15 min at room temperature, followed by centrifugation to recover the cells. After this, the cells were treated with 60–100% of EtOH stepwise and finally diluted in 20 µL of 100% of EtOH. After EtOH treatment, 2 µL of the cells were loaded onto a membrane filter (0.22 µm; Agela Technologies Inc., Wilmington, DE, USA) and imaging was done under the scanning electron microscope (JSM-6700F, JEOL, Tokyo, Japan) [21].

### 2.4. Fresh Agricultural Food (Lettuce) Application Test

Combination treatment of recombinant SAP8 endolysin and nisin was applied to actual food lettuce to confirm whether it exerts an effect on the growth inhibition of *S. aureus* with a modification of previously published method [22]. The lettuce was washed in water for 30 s and sterilized by immersion in 70% EtOH for 15 min. Afterwards, the lettuce was cut to an appropriate size (2 cm × 2 cm) on a clean bench and dried under UV treatment for 30 min. The overnight culture of 10^5^ CFU/mL cells of *S. aureus* KCTC 3881 strain was inoculated on lettuce sample and dried at 37 ℃ for 2 h. After drying, the lettuce sample treated with 10^5^ CFU/mL cells of *S. aureus* KCTC 3881 strain was immersed in 2 mL of a mixture of SAP8 endolysin (0.01 µM) and nisin (18 IU/mL) to examine their growth inhibition effects. 0.008% of sodium hypochlorite (NaClO) was also applied separately as positive control on lettuce samples inoculated with *S. aureus* KCTC 3881. The lettuce sample was left at room temperature for 1 h, aseptically recovered, put in a tube containing 0.1% peptone water, crushed, and vortexed. For viable cell count, 10 times serial dilutions were made, pouring 10 µL of each dilution on MRS agar plate in triplicate form. After incubation at 37 ℃ for 24 h, colonies were counted and determined CFU/mL using formula (total colonies × dilution factor × 100).

## 3. Results

### 3.1. Growth Inhibition of S. aureus KCTC 3881 with Recombinant SAP8 Endolysin

The antimicrobial activity of recombinant SAP8 endolysin was investigated against *S. aureus* KCTC 3881 in a dose-dependent manner after 3 h of incubation. The recombinant SAP8 endolysin showed antimicrobial activity against *S. aureus* KCTC 3881 at a concentration range from 0.001 µM to 1 µM. Typically, 5.93 log CFU/mL of *S. aureus* KCTC 3881 was gradually decreased by increasing the concentration of recombinant SAP8 endolysin. The cell count reduction was negligible in the presence of 0.001 µM of recombinant SAP8 endolysin, while an increase in the concentration to 0.01 µM, 0.1 µM, and 1 µM resulted in a notable reduction in the cell count after 3 h of incubation. The 5.93 log CFU/mL of *S. aureus* KCTC 3881 was reduced to 3.64 log CFU/mL after treatment with 1 µM of recombinant SAP8 endolysin (Figure 1 and Table 1).

### 3.2. Growth Inhibition of S. aureus KCTC 3881 Followed by Co-Treatment with Recombinant SAP8 Endolysin and Nisin

The antimicrobial activity of nisin and recombinant SAP8 endolysin were compared individually and in combination against *S. aureus* KCTC 3881. A decrease in the spectrum of viable cell counts after individual treatment with 9 IU/mL and 18 IU/mL of nisin and 0.01 µM of recombinant SAP8 endolysin was noticed, but the inhibition of *S. aureus* KCTC 3881 in response to the combination of different concentrations of nisin and 0.01 µM of recombinant SAP8 endolysin was identified to be much more effective. Typically, 0.01 µM of recombinant SAP8 endolysin was combined with 9 IU/mL and 18 IU/mL of nisin, and the inhibition of *S. aureus* KCTC 3881 was investigated. The viable cell counts of *S. aureus* KCTC 3881 after both the treatments reached 4.26 and 3.85 log CFU/mL from 5.90 log CFU/mL, respectively (Figure 2 and Table 1). Combination of 0.01 µM of recombinant SAP8 endolysin and 18 IU/mL of nisin completely inhibited the growth of *S. aureus* in resazurin assay. This combined effect observed in resazurin assay could be due to possible reasons such as long incubation time and decreasing the total number of *S. aureus* cells to 5 log CFU/mL.

### 3.3. Growth Inhibition of S. aureus KCTC 3881 on Lettuce

Combination effect of recombinant SAP8 endolysin and nisin was also assessed on fresh agricultural food lettuce against *S. aureus* KCTC 3881. 0.01 µM of recombinant SAP8 endolysin was combined with 18 IU/mL of nisin and applied against *S. aureus* KCTC 3881 on lettuce. In addition, 0.008% of sodium hypochlorite (NaClO) was also applied separately as positive control on lettuce samples inoculated with *S. aureus* KCTC 3881. The viable cell counts of *S. aureus* KCTC 3881 were decreased to zero after being treated with NaClO. However, the viable cell counts of *S. aureus* KCTC 3881 were gradually decreased after treatment with combination of 0.01 µM of recombinant SAP8 endolysin and 18 IU/mL of nisin as compared to control after 1 h of incubation. After 6 h of incubation, the viable cell count of *S. aureus* KCTC 3881 reached half of the original cell count (Figure 3).

Additionally, cellular destruction of *S. aureus* KCTC 3881 was investigated by scanning electron microscope after using 18 IU/mL of nisin and 0.01 µM of recombinant SAP8 endolysin compared with untreated cells of *S. aureus* KCTC 3881. The complete cellular destruction of *S. aureus* KCTC 3881 cells was observed after treatment as compared with the control (Figure 4).

## 4. Discussion

In this study, we have demonstrated that the successful treatment of pathogenic *S. aureus* KCTC 3881 depends on an essential combination between different concentrations of recombinant SAP8 endolysin and nisin. Based on the comparative treatment of *S. aureus* KCTC 3881 with recombinant SAP8 endolysin and nisin, a gradual decrease in the viable cell count of *S.*
*aureus* KCTC 3881 was observed, with an increase in the concentrations of recombinant SAP8 endolysin and/or nisin; this technique was not identified as a cost-effective approach. However, co-treatment of recombinant SAP8 endolysin and nisin against the strain led to a significant decrease in the viable cells.

Antibiotic-resistant pathogenic bacteria have become a rising concern worldwide with an increase in the risks of foodborne illness, human and zoonotic transmission, especially methicillin-resistant *S. aureus* (MRSA), which has been frequently reported due to consumption of contaminated milk, animal or human cross contamination, or through food chain such as during animal slaughtering, food processing, or consumption of contaminated dairy products [5,15,23,24,25]. *S. aureus* strains prevail as commensal opportunistic nasal microbiota in humans and animals, disruption in the linings of cutaneous and mucosal barriers cause *S. aureus* to get access to the underlying tissues and cause infection [26]. *S. aureus* strains form complex biofilm community in different hosts surrounded by extracellular polymeric substances (EPS). These EPS protect *S. aureus* community from antibacterial agents [10,27]. Besides this, *S. aureus* evolved independently to acquire SCC*mec* complex, executing *S. aureus* to cause resistance to β-lactam family members of antibiotics [28]. Twelve known different types of SCC*mec* have been found and among them type I, II, III, IV, V, and VI harbor genes of large SSC*mec* elements that confer resistance in MRSA [29]. All types of SCC*mec* carried *mecA* encode penicillin-binding protein 2a (PBP2a), which has very low affinity for β-lactam family members [30]. In several *S. aureus* strains from human and animals, *mecC* and *mecB* (*mecA* variants) were also identified [31,32]. The development of resistance to multiple antibiotics has urged scientists to discover new ways of treatment and bacteriophages have become a powerful tool to treat these bacteria [6,33,34]. The antipathogenic effects of the bacteriophages have been successfully reported in food, animals, aquaculture, crops, and humans [35,36]. Several studies have reported isolation, characterization, and investigation of the protective effects of phages from *S. aureus* [16,37]. Phage therapy largely depends on phage selection possessing desired characteristics such as high stability, absence of virulence genes, lytic activity, and wide host range [10]. Phage host range depends on tail fibers that help in recognition of viral receptors on the bacterial surface [10]. Phages developed different systems to be protected by host defense systems, as phages lack targeted sequences to escape from *S. aureus* restriction-modification systems [10]. However, in certain cases phage therapy is not recommended, as phages are able to transfer virulence or antibiotic resistance genes from one bacterium to another bacterium [38]. In the *Myoviridae* family from *Staphylococcus*, phages encode SceD-like transglycosidase genes that increase vancomycin resistance of MRSA [39]. Hussain et al. have studied about the marine *Vibrio* resistance mechanism against phages [12,13]. Therefore, endolysin genes from phages responsible for host lysis could be a novel approach to combat these risk factors. Additionally, nisin an antimicrobial peptide has been widely used against pathogenic bacteria [21,23]. Nisin control a wide range of Gram-positive bacteria by forming pores in bacterial cytoplasmic membrane and by inhibiting cell wall synthesis [10,40,41,42]. In previous studies, it was reported that phage could prevent growth of *S. aureus* only at high concentration like 10^9^ PFU/mL, but in combination with nisin (100 IU/mL), low concentration of phage (10^8^ PFU/mL) could prevent regrowth of *S. aureus* [10]. Several reasons were suggested behind this phenomenon, such as nisin might form pores in bacterial cytoplasmic membrane, lytic enzymes on phage tail acted synergistically with nisin to cause damage in bacterial membrane, nisin might rapidly decrease bacterial cell counts and eventually enhance phage efficacy, nisin might kill phage resistant cells. Our results also provide a new mechanistic basis for the treatment of infections caused by *S. aureus* by combination of recombinant SAP8 endolysin and nisin. We showed that increasing concentration of SAP8 endolysin and nisin led to a decrease in the count of viable cells of *S. aureus* KCTC 3881 (Figure 1 and Table 1); however, treatment with 0.01 µM of recombinant SAP8 endolysin with 9 IU/mL and 18 IU/mL of nisin led to a significant decrease in the viable cell counts (Figure 2 and Table 1). The viable cell counts of *S. aureus* KCTC 3881 in fresh agricultural food lettuce were also decreased after combination treatment of SAP8 endolysin and nisin, which also supported our hypothesis (Figure 3). The results were further confirmed using the scanning electron microscope under which the combination effect of 0.01 µM of recombinant SAP8 endolysin and 18 IU/mL of nisin led to the complete destruction of the *S. aureus* cells (Figure 4). Previously, combination treatment with endolysin LysH5 from bacteriophage and nisin was studied against *S. aureus* in pasteurized milk, and the synergistic effect showed results similar to ours after 6 h of incubation [43]. Our study reinforces and suggests the combination treatment with recombinant SAP8 endolysin and nisin as a potent alternative to combat the growth of pathogenic *S. aureus*.

## 5. Conclusions

Even though SAP8 endolysin and nisin are good candidates for controlling the pathogenic bacteria *S. aureus*, a combination of recombinant SAP8 endolysin and nisin might be an efficient way, and could be a strong alternative to antibiotics, to control the growth of *S. aureus*, including MRSA.

## Figures and Tables

**Figure 1 antibiotics-11-01185-f001:**
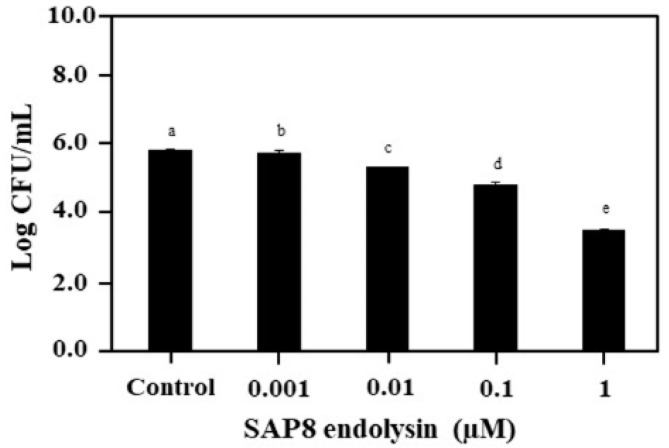
Growth inhibition of *Staphylococcus aureus* KCTC 3881 with recombinant SAP8 endolysin ranging from 0.001–1 µM. Data are provided as the mean value ± SD, measured in triplicate. Different letters present significant differences (*p* < 0.05).

**Figure 2 antibiotics-11-01185-f002:**
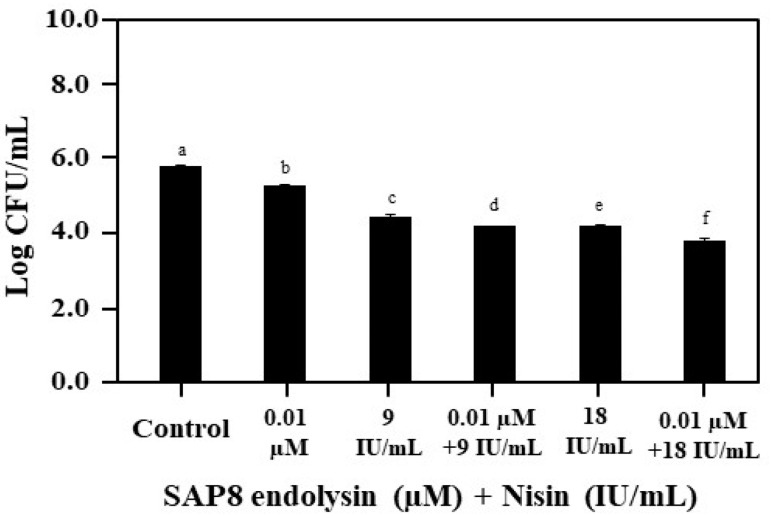
Growth inhibition of *Staphylococcus aureus* KCTC 3881 with recombinant SAP8 endolysin (µM) and nisin (IU/mL). Data are provided as the mean value ± SD, measured in triplicate. Different letters present significant differences (*p* < 0.05).

**Figure 3 antibiotics-11-01185-f003:**
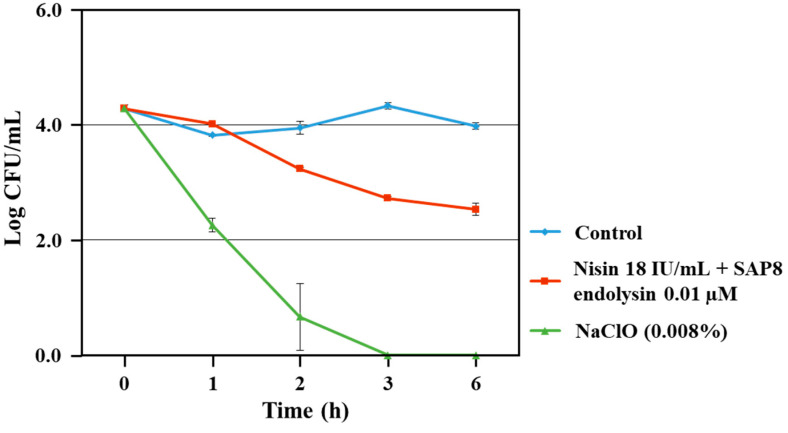
Growth inhibition of *Staphylococcus aureus* KCTC 3881 in lettuce by combination of SAP8 endolysin and nisin. Data are provided as the mean value ± SD, measured in triplicate.

**Figure 4 antibiotics-11-01185-f004:**
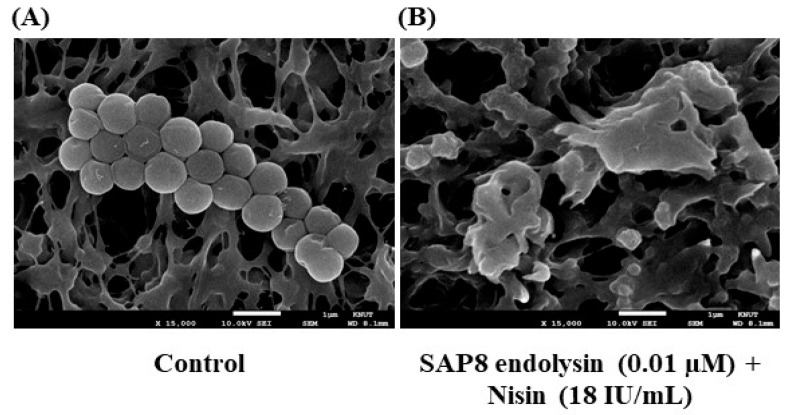
Scanning electron micrographs for comparison of the cellular destruction of *Staphylococcus aureus* KCTC 3881 of control (no treatment) (**A**) in the presence of the combination of 0.01 µM of recombinant SAP8 endolysin and 18 IU/mL of nisin (**B**) after incubation at 30 ℃ for 3 h.

**Table 1 antibiotics-11-01185-t001:** Growth inhibition of *Staphylococcus aureus* KCTC 3881 under different treatments.

Treatment	*S. aureus* (Log CFU/mL)
Control (DW)	5.90
Sap8 endolysin (µM)	
0.001 µM	5.83
0.01 µM	5.39
0.1 µM	4.88
1 µM	3.55
Nisin (IU/mL)	
9 IU/mL	4.53
18 IU/mL	4.26
SAP8 endolysin (µM) + Nisin (IU/mL)	
0.01 µM + 9 IU/mL	4.26
0.01 µM + 18 IU/mL	3.85

## Data Availability

Data are available in publicly accessible repository and within the article.

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
