# Peer review of "Combination Treatment for Inhibition of the Growth of *Staphylococcus aureus* with Recombinant SAP8 Endolysin and Nisin"

_antibiotics, 2022, doi:10.3390/antibiotics11091185_

Round 1

Reviewer 1 Report

The article "Combination treatment for inhibition of the growth of Staphylococcus aureus with recombinant SAP8 endolysin and nisin " is a bit interesting. This study aims to examine the combined effects of recombinant SAP8 endolysin from the bacteriophage SAP8 and commercial nisin in controlling the growth of S. aureus KCTC 3881. However, there are some important weaknesses in this work.

First of all, the authors should clarify the reason for choosing the bacteriophage SAP8 and commercial nisin to combat S. aureus at least in the introduction part. According to other studies, nisin has high antimicrobial activity against many different gram‐positive bacteria, while SAP8 was shown as S. aureus-specific phage. How far did the previous research about SAP8 and nisin on S. aureus go?

Secondly, it looks like that bacteriophage SAP8 and commercial nisin have synergistic antibacterial action on S. aureus, but the authors even didn’t determine the MIC for SAP8 and nisin on the strain.

Thirdly, the authors only selected one isolate of S. aureus as the subject to evaluate the effects of endolysin and nisin, while the isolate’s description was absent. Is the isolate MRSA or MSSA? Why did the authors choose this isolate? The inhibition effect might be strain-specific.

Therefore, I think the current work is not so useful addition to the literature in its current status.

Author Response

We appreciate for the very kind and excellent comments from the reviewers and editor. We have carefully revised the manuscript based on the comments and responded to all comments point-by-point. Please find an attached file.

Reviewer 2 Report

The manuscript entitled

 Combination treatment for inhibition of the growth of Staphylococcus aureus with recombinant SAP8 endolysin and nisinis discussing an interesting topic. Alternative approaches rather than antimicrobial treatment in human and veterinary practice are not optional any more.  However, the methodology and presenting of the results in addition to scientific soundness are unfortunately inadequate.

Experimental results are not adequately described, their interpretation is not reasonably described and the discussion has to be improved. I have some key comments, to mention:

Minor language spell check is required.

Abstract:

Lines 15: involved instead of involve

Line 20: Combined instead of combination

The authors have to state the breakpoint at which concentration of endolysin or nisin either separately or combined the S. aureus growth is totally inhibited.

Introduction:

Line 33: remains instead of remained

Line 34: remain instead of remained. Globally instead of among scientists.

Line 37: Please write always at first time Staphylococcus (S.) aureus instead of S. aureus.

Line 38: antimicrobials instead of antimicrobial

Material and Methods:

Lines 58 and 59: The recombinant DNA for SAP8 endolysin was previously constructed and ex-58 pressed in Escherichia coli strain and the recombinant protein was named LysSAP8. Please state whether you have followed the same protocol or this is just an information.

Line 61: S. aureus KCTC 3881 strain. Please mention the background for choosing this strain exactly and the source of the isolate.

Line 64: distilled water and diluted to attain 108 CFU/mL of cells approximately. Please mention how could you confirm the CFU

Line 72 and 73: nisin (9 or 18 IU/mL; Galacin Nisin 101, Galactic, Belgium). Please mention the concentration you have used.

Lines 83 and 84:

Here, the S. aureus KCTC 3881 strain cultured overnight was inoculated at 5 83 Log CFU/ml, dried at 37℃ for 2 h, and used for the experiment. That is, the pre-treated 84 dried lettuce was immersed in 2 mL of a mixture of SAP8 endolysin (0.01 uM) and nisin 85 (18 IU/mL), Please rephrase it is not clear

Discussion:

The discussion should not be only a comparison between results. Authors have to state logic justifications for their results in relation to the way of sample collection, geographical variation, climate change and sex variation.

Authors also have to discuss the safety of both Zinc oxide nanoparticles when administered orally in human and animal. That was very important also in the methodology.

Hereby, I recommend major changes before accepting to publish the manuscript.

Results:

Line 325: Table 2 needs to be redesigned.

References:

Line 87: and the viable cell count of the strain was measured after suspension. Please mention how could you count the viable cells

Lines 91-97: Please include the reference for methodology and rephrase the whole paragraph to be more clear

Results

Lines 123 and 124: of viable cell counts was noticed against 9 IU/mL and 18 IU/mL of nisin and 0.01μM of recombinant SAP8 endolysin. Please make it clear in the text if you used each of them separately or together.

Line 126: lysin was identified to be much more effective. How much decrease in log?

The authors must mention why did not they use the combination of 9 and 18 IU/mL Nisin in combination with 1µM endolysin as it was clear that the combination of Nisin both concentrations with 0.1 µM endolysin did not completely inhibit the growth of S. aureus. In this case, the authors must use the higher concentration to prove whether it can completely inhibit the growth or not. A breakpoint has to be determined.

It would be much easier if the authors could include a table with all data of different concentration of both Nisin and Endolysin separately and combined and the result of each application including log decrease in each case.

Discussion

Line 185: what is the reason for choosing milk specifically as a source of MRSA? Then you have to illustrate all other sources with references.

The authors had to mention three main points; the pathogenicity of MRSA in human and animals, mechanism of methicillin resistance in S. aureus and more importantly the mechanism of action of both nisin and endolysin as antibacterials against S. aureus. The authors also had to talk about results of similar studies with these applications and the obtained results to compare their results with. Then they could notice if their application was done in the best way.

I believe that although the soundness and novelty of the study, many important perspectives are missed in the study.

I would recommend major revisions before publishing the manuscript.

Author Response

(The authors gave the same response as above.)

Round 2

Reviewer 1 Report

Thanks for the authors' explanation and update. However, since the authors have already tested activity of recombinant SAP8 endolysin against several strains of MRSA, I sincerely expect the author could show the result and explain why they chose S. aureus strain KCTC 3881 as a model strain for further studies and if the inhibition effect is MRSA specific.

Author Response

We really appreciate for the valuable comment and added some mention for the reason why we chose KCTC 3881 strain. Thank you so much.

We appreciate for the excellent comment from the reviewer.

Responses to reviewer

  1. line 71-76, why they chose S. aureus strain KCTC 3881 as a model strain for further studies and if the inhibition effect is MRSA specific; We have mentioned the reason in the manuscript for choosing aureus KCTC 3881 as a model strain, actually in our previous study, we have tested crude nisin from L. lactis strain against several S. aureus strains and found S. aureus KCTC 3881 as the most susceptible strain among all strains and was chosen as a model strain in this study.

Additionally, in another study, recombinant SAP8 endolysin was tested against 75 strains of S. aureus including MRSA strains and recombinant SAP8 showed activity against 73 strains of S. aureus that showed recombinant SAP8 is not strain specific.

Reviewer 2 Report

I hereby accept the modificatons added by the authors, and accept the current version to be published.

Author Response

Thank you so much for the valuable evaluation.